# A Functional Network Model of the Metastasis Suppressor PEBP1/RKIP and Its Regulators in Breast Cancer Cells

**DOI:** 10.3390/cancers13236098

**Published:** 2021-12-03

**Authors:** Mahmoud Ahmed, Trang Huyen Lai, Wanil Kim, Deok Ryong Kim

**Affiliations:** Department of Biochemistry and Convergence Medical Science, Institute of Health Sciences, Gyeongsang National University College of Medicine, Jinju 527-27, Korea; mahshaaban@gnu.ac.kr (M.A.); tranghuyen@gnu.ac.kr (T.H.L.); wikim@gnu.ac.kr (W.K.)

**Keywords:** metastasis, breast cancer, reverse-causal-reasoning, RKIP/PEBP1

## Abstract

**Simple Summary:**

We extracted known interactions between metastasis suppressors and their regulators from scientific studies. Next, we used publically available data of drug treatments to identify which of them potentially perturbs these interactions. Finally, we studied the effect of several of these drugs on a particular metastasis suppressor called RKIP and found that our model accurately predicted its regulation in breast cancer cells. Our approach can discover alternative mechanisms of existing cancer drugs and repurpose them in different disease types.

**Abstract:**

Drug screening strategies focus on quantifying the phenotypic effects of different compounds on biological systems. High-throughput technologies have the potential to understand further the mechanisms by which these drugs produce the desired outcome. Reverse causal reasoning integrates existing biological knowledge and measurements of gene and protein abundances to infer their function. This approach can be employed to appraise the existing biological knowledge and data to prioritize targets for cancer therapies. We applied text mining and a manual literature search to extract known interactions between several metastasis suppressors and their regulators. We then identified the relevant interactions in the breast cancer cell line MCF7 using a knockdown dataset. We finally adopted a reverse causal reasoning approach to evaluate and prioritize pathways that are most consistent and responsive to drugs that inhibit cell growth. We evaluated this model in terms of agreement with the observations under treatment of several drugs that produced growth inhibition of cancer cell lines. In particular, we suggested that the metastasis suppressor PEBP1/RKIP is on the receiving end of two significant regulatory mechanisms. One involves RELA (transcription factor p65) and SNAI1, which were previously reported to inhibit PEBP1. The other involves the estrogen receptor (ESR1), which induces PEBP1 through the kinase NME1. Our model was derived in the specific context of breast cancer, but the observed responses to drug treatments were consistent in other cell lines. We further validated some of the predicted regulatory links in the breast cancer cell line MCF7 experimentally and highlighted the points of uncertainty in our model. To summarize, our model was consistent with the observed changes in activity with drug perturbations. In particular, two pathways, including PEBP1, were highly responsive and would be likely targets for intervention.

## 1. Introduction

Several approaches for screening and identifying bioactive compounds exist and are being used to find effective remedies for cancer. Most strategies focus on quantifying the phenotypic effects of different biological models [1]. High-throughput technologies like massive parallel sequencing and mass spectrometry have the potential to augment these strategies by explaining the effect of panels of drugs on biological entities such as gene expression and protein abundance. Reverse causal reasoning enables the use of quantitative measurements of entities in a given pathway to make claims about their functions [2,3]. This method integrates existing biological knowledge and data of gene and protein abundance to infer their function [4]. In addition to taking advantage of prior knowledge, this approach utilizes the network structure, which is key in biological pathways and avoids making assumptions about equating gene expression with protein abundance or activity [5,6].

Breast cancer is the most prevalent type of cancer among women, and the vast majority of deaths are due to metastasis [7,8]. Cancer metastasis is a complex process that involves disseminating cancer cells from primary solid tumors to distant parts of the body, where they form secondary tumors. Effective cancer therapies require an understanding of the underlying mechanisms and regulatory pathways of genes involved in metastasis. A group of genes, known as metastasis suppressor genes (MSG), prevents the formation of overt metastases [9]. Although multiple MSGs have been discovered, the detailed mechanism by which they overcome metastasis is not well understood. One of the metastasis suppressors is phosphatidylethanolamine binding protein (PEBP1), also known as RAF kinase inhibitory protein (RKIP) [10]. PEBP1 inhibits cancer metastasis at different stages, including epithelial to mesenchymal transition (EMT), migration, and invasion [11,12,13]. Low expression of PEBP1 is associated with cancer metastasis and poor prognosis [14].

Here, we attempted to integrate two sources of data in the form of biological knowledge and gene expression profiles to build a functional model of metastasis in breast cancer. We used a reverse causal reasoning approach to infer the function of crucial metastatic proteins from the perturbation of expression of other gene products known to interact with them. We evaluated the model for agreement in the direction of change with drug treatments that inhibit the growth of the cancer cells. Finally, we prioritized a few key pathways involving the anti-metastasis PEBP1 for further testing.

## 2. Materials and Methods

### 2.1. Protein-Protein Interactions, Literature Search, and Biological Expression Language

We used the String database to identify protein–protein interactions (PPI) among a list of seventeen MSGs (Table 1), and eleven transcription factors (TFs) (Table 2) [15]. Most of the retrieved interactions were based on text mining, where two or more entries co-occur in published articles. We surveyed the linked studies for evidence to establish causal relations between the entities. We encoded these interactions in the biological expression language (BEL) and assigned at least one reference to each (Table 3) [16]. In BEL, two entities (e.g., r, mRNA or p, protein) interact when a subject produces an effect (increase, decrease, directlyIncrease or directlyDecrease) on the object. We used these interactions to build the functional layer of the metastatic suppressors and their regulators.

### 2.2. Knockdown of Metastasis Suppressors and Transcription Factors in MCF7

We used two datasets where each MSG and TF was knocked down in the MCF7 breast cancer cell line. The first is a dataset that consists of 77 microarray samples [28]. Seventeen MSGs were knocked down (siRNA) separately in MCF7 (Table 1). The control set consisted of wild-type (0 h) and cells treated with scramble siRNA at 48 or 96 h. We obtained the processed data using GEOquery [29]. We mapped the probe intensities to gene symbols and applied differential expression between knockdown and control using limma [30]. We compiled a second dataset from the KnockTF database for the eleven TFs knockdown in the same cell line (Table 2) [31]. KnockTF curates gene expression profiles of TFs knockdown from the gene expression omnibus (GEO) and presents the results in the form of differential expression output (fold-change and *p*-values).

### 2.3. Pharmacological Perturbations of Breast Cancer Cell Lines

Two sources of publicly available drug perturbation data were used in this study. The first is growth inhibition (GRmax) data of drug perturbations (*n* = 35) in breast cancer cell lines (*n* = 71, including MCF7) [32]. Adenosine triphosphate content was measured as a proxy for the cell count in drug treated cell cultures relative to DSMO-treated controls. We included in the analysis only the drugs with at least one effective dose. This dataset was used as evidence for the efficacy of these drugs in producing phenotypic effects on cancer cells. The second data source is the library of integrated network-based cellular signatures (LINCS), which contains gene expression profiles of multiple cells under different types of perturbations, including compounds, gene overexpression, knockdown, or knockouts [33]. Several cell lines were profiled for gene expression using the L1000 technology, which measures the abundance of 1000 landmark genes and infers the expression of the rest of the genes. We used a subset of drug perturbations (*n* = 35, same set of drugs as above) in cancer cell lines (*n* = 67) that were also surveyed for drug sensitivity. We obtained the expression profiles of all genes (level 3 of the LINCS data) using Slinky [34].

### 2.4. Network Perturbation Amplitude (NPA)

Network perturbation amplitude (NPA) models the perturbation of a causal network as the changes in the expression of connected downstream nodes [4]. This method assumes that the downstream gene expression reflects the function of the entities in a biological pathway (e.g., the activity of a given TF). The input to this method is a two-layer network model of the pathway of interest and expression profiles of two or more conditions. First, the functional layer (backbone) encodes the causal relations between the entities of interest. The second is a transcription layer (terminal) with all the nodes downstream from each node in the backbone. We used the gene expression profiles from LINCS pharmacological perturbations to score the model. NPA is an R package that efficiently implements the method with the same name [35].

**Table 3 cancers-13-06098-t003:** Interactions of metastasis suppressor genes and their transcription factors.

Subject	Object	Ref.	Interaction
Interactions between metastasis suppressors			
CD44	CASP8	[36]	p(CD44) decreases act(p(CASP8))
CD82	CD44	[37]	p(CD82) decreases p(CD44)
CD82	BRMS1	[38]	p(CD82) increases r(BRMS1)
CD82	CDH1	[39]	p(CD82) increases r(CDH1)
CDH1	CASP8	[40]	p(CDH1) increases act(p(CASP8))
GSN	CDH2	[41]	p(GSN) increases r(CDH2)
MAP2K3	MAPK14	[42]	act(comp(p(MAP2K6), p(MAP2K3)), ma(kin)) directlyIncreases p(MAPK14)
MAP2K4	MAP2K7	[43]	p(MAP2K4) decreases act(p(MAP2K7), ma(kin))
MAP2K4	MAPK14	[42]	p(MAP2K4) directlyIncreases act(p(:MAPK14), ma(kin))
MAP2K4	CASP8	[44]	p(MAPK14) increases p(CASP8)
MAP2K6	MAPK14	[42]	p(MAP2K6) directlyIncreases p(MAPK14)
MAP2K7	CDH2	[45]	p(MAP2K7) increases r(CDH2)
MAPK14	CDH1	[46]	p(MAPK14) increases p(CDH1)
MAPK14	CASP8	[47]	act(p(MAPK14), ma(kin)) directlyDecreases p(CASP8)
MAPK14	SAP1A	[48]	act(p(MAPK14), ma(kin)) directlyIncreases p(ets-Domain Protein Elk-4)
MAPK14	RUNX2	[49]	p(MAPK14) increases r(RUNX2)
NME1	BRMS1	[50]	p(NME1) decreases r(BRMS1)
NME1	AKAP12	[50]	p(NME1) increases r(AKAP12)
NME1	PEBP1	[51]	r(NME1) increases r(PEBP1)
PEBP1	MAP2K6	[52]	p(PEBP1) increases act(p(MAP2K6))
PEBP1	MAP2K3	[52]	p(PEBP1) increases act(p(MAP2K3))
PEBP1	MAPK14	[52]	p(PEBP1) increases act(p(MAPK14))
PEBP1	CDH1	[53]	p(PEBP1) increases r(CDH1)
RUNX2	CDH2	[54]	p(RUNX2) decreases r(CDH2)
RUNX2	CDH1	[54]	p(RUNX2) increases r(CDH1)
TGFB1	CDH1	[55]	p(TGFB1) decreases r(CDH1)
TGFB1	CDH2	[56]	p(TGFB1) decreases r(CDH2)
Interactions between metastasis suppressors and regulators			
CDH1	YBX1	[57]	p(CDH1) decreases p(YBX1)
CDH1	FOXM1	[58]	p(CDH1) decreases p(FOXM1)
ESR1	CDH2	[59]	p(ESR1) decreases r(CDH2)
ESR1	CDH1	[60]	p(ESR1) directlyIncreases r(CDH1)
ESR1	BRMS1	[61]	p(HGNC:ESR1) directlyIncreases r(HGNC:BRMS1)
ESR1	MTA3	[62]	p(ESR1) increases r(MTA3)
ESR1	RUNX2	[63]	p(ESR1) increases r(RUNX2)
FOS	CDH1	[55]	p(FOS) decreases r(CDH1)
FOS	RUNX2	[64]	p(FOS) decreases r(RUNX2)
FOS	TGFB1	[55]	p(FOS) directlyDecreases r(TGFB1)
FOS	CD44	[65]	p(FOS) increases r(CD44)
GATA3	CD44	[66]	r(GATA3) directlyIncreases r(CD44)
GATA3	CD44	[66]	r(GATA3) directlyIncreases r(CD44)
GATA3	CD44	[66]	p(GATA3) increases r(CD44)
GATA3	CD44	[66]	p(GATA3) increases r(CD44)
HIF1A	CD44	[67]	p(HIF1A) increases r(CD44)
MAP2K4	FOS	[68]	p(MAP2K4) increases p(FOS)
MAPK14	FOS	[48]	act(p(MAPK14), ma(kin)) increases p(FOS)
MAPK14	GATA3	[69]	p(MAPK14) increases act(p(GATA3))
MTA3	CDH2	[62]	p(MTA3) decreases r(CDH2)
MTA3	CDH1	[62]	p(MTA3) increases r(CDH1)
RELA	BRMS1	[70]	p(RELA) directlyDecreases r(BRMS1)
RUNX2	HIF1A	[71]	p(RUNX2) increases act(p(HIF1A))
RUNX2	SPDEF	[72]	p(RUNX2) directlyDecreases r(SPDEF)
SAP1A	FOS	[48]	p(ets-Domain Protein Elk-4) directlyIncreases r(FOS)
SATB1	NME1	[73]	p(SATB1) directlyDecreases r(NME1)
SATB1	BRMS1	[73]	p(SATB1) directlyDecreases r(BRMS1)
SATB1	CD82	[73]	p(SATB1) directlyDecreases r(CD82)
SATB1	CDH1	[73]	p(SATB1) directlyDecreases r(CDH1)
SNAI1	CDH1	[74]	p(SNAI1) directlyDecreases r(CDH1)
SNAI1	PEBP1	[75]	p(SNAI1) directlyDecreases r(PEBP1)
SNAI1	CASP8	[76]	p(SNAI1) decreases act(p(CASP8))
TFAP2C	CD44	[77]	p(TFAP2C) directlyDecreases r(CD44)
YBX1	CD44	[78]	p(YBX1) directlyIncreases r(CD44)
Interactions between transcription factors			
ESR1	HIF1A	[79]	p(ESR1) directlyIncreases r(HIF1A)
ESR1	GATA3	[20]	p(GATA3) increases p(ESR1)
FOXM1	GATA3	[80]	p(FOXM1) directlyDecreases r(GATA3)
MTA3	SNAI1	[81]	p(MTA3) directlyDecreases r(SNAI1)
NR5A2	ESR1	[82]	p(NR5A2) directlyIncreases r(ESR1)
RARA	FOS	[83]	p(RARA) decreases act(p(FOS))
RELA	SNAI1	[84]	p(RELA) directlyIncreases r(SNAI1)
TFAP2C	ESR1	[85]	p(TFAP2C) directlyIncreases r(ESR1)
TNFSF10	CASP8	[86]	p(TNFSF10) increases p(CASP8)

### 2.5. Measures of Agreement

We defined two notions of agreement between expectations and observations of the effects of drug perturbations on the metastasis network model. For every node (*u*), we created a graph of all *n* nodes connected to it by an edge (*e*) (subnetwork). Given a binarized perturbation coefficient of drug treatment, we formed an expectation by multiplying +1 or −1 with the direction of interaction +1 or −1 for every edge, u×e. We compared the expected sign to the observed perturbation coefficient of the downstream nodes ($x^{\prime }$). The average for every subnetwork we termed a **concordance** rate. We similarly computed an agreement estimate for all possible directed graphs that lead to a particular node (path). The difference from above is that the upstream node is not fixed, but changes as we move one edge down the path. The average agreement between the observed and expected effect for every path we termed a **coherence** rate. The following formula shows how the agreement rates were calculated.
$$\frac{1}{n}\sum_{i=1}^{n}x_{i}\;x=\left\{\begin{array}{cc} 1, & \mathrm{if}\;u.e=x^{\prime }\\ 0, & \mathrm{otherwise} \end{array}\right.u=\left\{\begin{array}{cc} 1, & \mathrm{if}\;u\;\mathrm{is}\;\mathrm{activated}\\ -1, & \mathrm{if}\;u\;\mathrm{is}\;\mathrm{repressed} \end{array}\right.$$
where *x* is a component connecting a node *u* to a downstream node by an edge *e* and $x^{\prime }$ is the observed effect of drug perturbation on that downstream node in the component of the subnetworks (*concordance*) or paths (*coherence*).

In both cases, the agreement was expressed in terms of Cohen’s κ [87]. This is a value between −1 (worse) and 1 (better) while taking into account the expected agreement by chance. The following shows how to calculate it.
κ=(observedagreement−expectedagreement)/(1−expectedagreement)

Finally, we compared the probability distribution of the agreement measures to randomly generated values using the Kolmogorov–Smirnov (KS) test. $D^{+}$ is the maximum distance between the cumulative distribution functions (ECDF).

### 2.6. Reagents and Drugs

Reagents and drugs utilized in this study were purchased as follows: RPMI-1640 media (11875-119), fetal bovine serum (FBS; 16000-044), Epirubicin hydrochloride (CAS 56390-09-1), Vorinostat/SAHA (CAS 149647-78-9), Methotrexate hydrate (CAS133073-73-1), Cisplatin (CAS 15663-27-1), Sorafenib (CAS 284461-73-0) from Sigma-Aldrich (St. Louis, MO, USA); Imatinib (CAS 220127-57-1) from STEMCELL Technologies Inc. (Vancouver, BC, Canada); Trizol reagent (15596026) from Invitrogen (Carlsbad, CA, USA); DNase I Solution (1 unit/μL), RNase-free (89836), and RevertAid First Strand cDNA Synthesis Kit from Thermo Scientific (Waltham, MA, USA); amfiSure qGreen Q-PCR Master Mix(2X), Without ROX (Q5600-005) from GenDEPOT (Katy, TX, USA).

### 2.7. Cell Culture and Drugs Treatment

MCF-7 breast cancer cells were cultured in RPMI-1640 supplemented with 10% Fetal Bovine Serum (GIBCO) and 100 μg/mL streptomycin and incubated in a 37 ${}^{\circ }$C humidified atmosphere containing 5% CO${}_{2}$. For drug treatment, cells ($3\times 10^{5}$ cells) were plated in a 6-well plate and further incubated for 24 h. Then, cells were treated with indicated drugs: Epirubicin, Vorinostat, Methotrexate, Cisplatin, Sorafenib, and Imatinib at 1.5 μM concentration and collected for assays after 24 h of treatment.

### 2.8. RNA Extraction, and RT-qPCR

Total RNAs were extracted from MCF7 breast cancer cells using Trizol reagent according to the manufacture’s manual. Prior to cDNA synthesis, RNA samples were treated with DNase I solution to remove trace amounts of DNA. The prepared RNA samples were then used as templates for reverse transcriptase to generate First-strand cDNA using Thermo Scientific RevertAid First Strand cDNA Synthesis Kit. The first-strand cDNA synthesis products were used directly in qPCR using the amfiSure qGreen Q-PCR Master Mix kit. Primers of target genes used in the assay are presented in Table 4. The expression of the five genes was quantified in the treated samples relative to the control gene GAPDH and the control condition of the DMSO treatment. $\Delta \Delta C_{t}$ model was applied using the pcr R package [88]. Student *t*-test was used to compare the relative expression in each treatment to the control DMSO treated cells. *p*-values < 0.05 were considered significant. Experiments were performed in five or more replicates.

### 2.9. Software Environment and Reproducibility

This analysis was performed in R and using Bioconductor packages [89,90]. The software environment was packaged and distributed as a Docker image (https://hub.docker.com/r/bcmslab/antimetastatic, accessed on 29 November 2021). The code to run the analysis and reproduce the figures and tables in this manuscript is available as open-source (GPL-3) (https://github.com/BCMSLab/antimetastatic, accessed on 29 November 2021).

## 3. Results

### 3.1. A Workflow for Building a Network of the Metastasis Suppressors and Their Regulators

In this study, we built and evaluated a network model of MSGs and their transcriptional regulators in breast cancer (Figure 1A). First, we identified PPIs of seventeen MSG and eleven TFs in the String database. Then, we vetted and supplemented every link with information from the relevant literature to determine the type of entity, direction of interaction, and evidence. We added a few interactions that were not present in the PPIs.

Since these causal links originated from experiments in different organisms and conditions, we further filtered the model to keep only the interactions relevant to the context of breast cancer. We obtained and analyzed the knockdown datasets of each of the input genes and regulators in the MCF7 cell line. We resolved the conflicts between the reported interactions and fold-change from the knockdown data by prioritizing the context of the interactions, the strength of the evidence, and the effect size (Figure 1B). We used differential expression to identify possible links between the nodes that were not previously reported and add them to the network. We compiled the results were in a *context-specific* metastasis model using BEL. The full list of included regulatory links is reported in Table 3 into three categories: within MSGs, within TFs, and in between the two groups of gene products.

Finally, we evaluated the context-specific model using gene expression and growth rate data in MCF7 treated with different compounds. The context-specific interactions served as a backbone to a transcript layer derived from the knockdown data to form a two-layer network. We scored the model using NPA under different conditions. We then compared the effect of the drugs on the entire network to their impact on the growth rate. Next, and given the expected effect of drugs on each node, we compared this predicted change on the downstream nodes to the observed using two different measures of agreement (Figure 1C,D). Moreover, we constructed a reliable model of highly consistent paths for further exploration.

### 3.2. Identifying Possible Interactions of Metastasis Suppressors Using Text Mining of the Literature

We took as a starting point the known protein–protein interactions between seventeen MSGs (Table 1) and their eleven TFs (Table 2). The initial query on the String database resulted in 93 interactions. Most interactions were based on text mining or the proximity of terms (gene products) in the text of one or more published study. We manually examined the provided evidence for each interaction. We also filtered, and supplemented with information from related studies. The interactions were coded in BEL and were used as the functional layer for the NPA analysis (Table 3). Each pair of interacting gene products were considered as a subject and an object. The type and direction of the relation between the two entities was represented according to the BEL standards.

### 3.3. Contextualizing Metastasis Suppressor Interactions in Breast Cancer Cells

Inevitably, conflicts arise between the relations suggested in the literature and the manifest changes of knocking down some genes on the expression of others. For example, not all interactions are relevant in the biology of breast cancer because they come from studies of different conditions. Additionally, not all interactions are reported in previous studies. We used a data-driven approach to specify the associations that are most relevant and augment the network with interactions based on changes in gene expression. The efficiency of the knockdown was verified by checking the expression of the corresponding genes (Figure 2, bottom). Most knocked down genes had a significant fold-change (log${}_{2}$ FC < −1 and *p*-value < 0.05) when compared to the control cells. Similarly, a wide-ranging change (between −4 and 2 log${}_{2}$ FC) in expression in the metastasis suppressors and regulators resulted from the knockdown, in most cases (Figure 2, top).

We compared the observed fold-change to the directions of previously curated interactions. We kept the ones that had no evidence to the contrary as a result of the knockdown of the subject. We removed the links with a significant (absolute log${}_{2}$ FC > 1 and *p*-value < 0.05) change in the opposite direction of change when the upstream node was knocked down. Finally, we coded the significant effects of the knockdown of one node (subject) on others (object) as an interaction between the mRNA to directly increase or decrease. The resulting network contained 78 edges between the input 35 unique nodes (Table 3). The recorded interactions included relations between the coding RNA, protein, and protein activities. That direction of interaction was either positive (increase/directlyIncrease) or negative (decrease/directlyDecrease) (Figure 3A).

TFs seem to have extensive interactions, reflecting the hierarchical nature of the regulator networks reported in other studies. Only a few nodes in the network had many edges, and most nodes had, on average, only two edges (Figure 3B, top). The highly connected nodes were either TFs that received signals from other regulators, key nodes in signaling pathways, or effector nodes that are tightly regulated. Examples include FOS and ESR1 as TFs, MAP Kinase subunits as signaling nodes, or CDH1 as an effector. In addition to filtering in the relevant interactions, we used the knockdown dataset to build a transcript layer for the functional model. The numbers of nodes connected to those in the functional layer were more normally distributed. Similarly, the distribution of up and down-regulation is normal but down-regulated nodes were more frequent (Figure 3B, bottom).

### 3.4. Evaluating the Metastasis Model Using Drug Perturbation Data

To evaluate the relevance of the metastasis network, we used a dataset of growth inhibition rates and gene expression profiles under drug perturbations. In this dataset, the growth rate was quantified in breast cancer cell lines (*n* = 71), including MCF7. Cell cultures were treated with several compounds (*n* = 35) and compared to the baseline replication rate. We included in the analysis only the drugs with at least one effective dose in terms of significantly (GRmax < 1) inhibiting cell growth (Figure 4A). Drug responses in most cell lines were correlated with the response in MCF7 (Figure 4B). NPA infers the activity of the nodes in the functional layer of the metastasis network using a transcript layer derived from the knockdown dataset. This method returns a perturbation coefficient that indicates the magnitude and direction of changes (−1, inhibition, and 1, activation) for the network as a whole and for each node in the graph (Figure 4C). The perturbation coefficients ranged from −0.1 to 0.2, with most nodes being significantly (confidence intervals not including zero) perturbed at least once. The perturbation amplitudes of 67 cancer cell lines correlated well with the NPA value in MCF7 (Figure 4D).

Next, we divided the metastasis graph into smaller modules and evaluated the agreement between the expected direction of change and the observed coefficients for each perturbation. We considered every node as an upstream connected to one or more nodes by one edge (*subnetwork*). The effect of the drugs on the upstream node was binarized (activation > 0, repression < 0) and used to infer the expected impact on the downstream nodes. We compared the expectations to the observed perturbation coefficients of the downstream node after similar binarization (*concordance*; see Section 2).

Higher concordance supports the consistency of the suggested model. Cohen’s κ was employed to evaluate the agreement between expectations and observation (κ > 0, better and κ < 0, worse), taking into account the chance agreement (50%). The average concordance of each drug was higher (D${}^{+}$ = 2.9 and *p*-value < 0.0001 in KS test) than expected by chance alone (Figure 5A). The network accurately predicted the direction of drug perturbations. Averaging for each node, a high concordance (D${}^{+}$ = 2.6 and *p*-value < 0.0001) was also observed (Figure 5B). The correlation between the concordance of the subgraph and the effect of the drug on its source node, a potential source of bias, was very weak (r < −0.02 and *p*-value > 0.9). We observed higher than chance (D${}^{+}$ = 2.47 and *p*-value < 0.0001) of consistency in other cell lines treated with the same set of drugs (Figure 5C).

### 3.5. Constructing a Model of PEBP1 and Its Interaction with Other Metastasis Suppressors

To demonstrate the utility of this model, we used the same perturbation data to prioritize key nodes and consistent edges between them. We isolated all paths in the graph leading to PEPB1 (*n* = 11). *Paths* are all interactions in the directed graph that end with the node corresponding to a protein. We multiplied the binarized perturbation coefficients of the upstream nodes by the sign of the edges between the nodes in every path. We compared these expectations were the observed perturbation coefficients on the downstream nodes of the paths (*coherence*; see Section 2). Not all drugs produced coherent effects on PEBP1 pathways, but the agreement was bigger than to be by chance (D${}^{+}$ = 2.44 and *p*-value < 0.0001 in KS test) (Figure 6A). Most paths were coherent (D${}^{+}$ = 1.45 and *p*-value < 0.0001) indicating the strong relevance of these pathways/interactions (Figure 6B). The main sources of bias (upstream coefficients) were not strongly relevant (r < −0.1 and *p*-value > 0.7). The observed directions of interaction were also consistent (D${}^{+}$ = 3.01 and *p*-value < 0.0001) with predictions across different cancer cell lines when treated with the same drugs (Figure 6C).

We used the top five coherent (coherence > 0.6) PEBP1 paths to highlight interactions that are strongly relevant to breast cancer cell metastasis and are highly responsive to treatment with different drugs (Figure 7A). We constructed a model of regulatory interactions between PEBP1 and its upstream regulators with the top five coherent paths represented by dash-lines (Figure 7B). Interestingly, we found that the pathways flowed in two main directions to enhance or suppress PEBP1 activity—the upstream proteins of PEBP1 function as TFs/regulators. One involves RELA (p65) and SNAI1, which were previously reported to inhibit PEBP1 transcription. The other involves the estrogen receptor (ESR1), which induces PEBP1 through the kinase NME1, which to our knowledge has not been reported before.

### 3.6. Validating the Model Predictions

To test the validity of the predictive model of PEBP1/RKIP regulation, we selected several drugs that activate or repress PEBP1 and tested their effect on gene expression in MCF7. Indeed, four out of 6 drugs conformed to the expected direction of regulation. Epirubicin induced the expression of PEBP1 while Sorafenib, Cisplatin and Imatinib repressed it (Figure 8A). One repressor (Sorafenib) seems to produce its effect on PEBP1 through either of the two suggested pathways (Figure 8B). Sorafenib treatment induced RELA and SNAI1, which lowered PEBP1 and repressed ESR1 and NME1, which activated it. The effect size on NME1 was small and not significant at a cuttoff of p=0.05. By contrast, Cisplatin and Imatinib repressed PEBP1 through one or the other regulatory pathways. Cisplatin induced RELA and SNAI1 which inhibited PEBP1 (Figure 8D), while Imatinib repressed PEPB1 through ESR1 and NME1 inhibition (Figure 8D). The activator, Epirubicin, induced the expression of PEBP1 through activating NME1 (Figure 8E). Our model predicted the direction of regulation of PEBP1 and the pathway by which this regulation was achieved (Figure 8F).

## 4. Discussion

We used text mining datasets and a manual literature search to extract evidence for possible interactions between several metastasis suppressors and their regulators. We filtered these interactions in the context of breast cancer using a knockdown dataset in the cell line MCF7. The resulting interactions were coded in BEL to build a functional metastasis model. Finally, we used a reverse causal reasoning approach to evaluate and prioritize these interactions and extract pathways that are most consistent with drug treatments that inhibit cell growth.

High-throughput technologies such as massive parallel sequencing and mass spectrometry produce simultaneous measurements on many biological entities. Interpreting these measurements requires expert biological knowledge. An approach to formalize the use of knowledge and data is to use causal reverse reasoning [4]. Here, we employ such an approach to infer the function of metastatic suppressors using curated causal links and the perturbation data. The implicit advantage of this approach is also to make claims about the role of gene products from measurements of gene expression. Castro and colleagues used a similar approach to construct a model of inflammatory bowel disease (IBD). They were able show that three pathways that contribute to the disease were regulated differently in the two subtypes of IBD [91]. Another study identified a mechanism the mediates the difference between the subtypes in response to environmental exposure such as cigarette smoke [92].

Studies based on text mining or manual curation alone can suffer from several drawbacks. Some of the identified interactions can only be relevant in a particular biological or pathological context. In addition, inconsistencies arise with changing the context of the interactions. Finally, inevitable biases exist in the literature toward studying specific pathways or models. We attempted to address these issues in our study with a number of strategies, which we will discuss next.

We took all possible interactions as defined by text mining of relevant studies and manual curation as a prior (Figure 1A). Next, we evaluated the relevance and consistency of these links. Interactions that were not contradicted by the knockdown of their corresponding nodes are likely to be present in this context (Figure 1B). This filtering step increases the likelihood of the reported links being relevant. Similarly, consistency in the direction of change with drug perturbations increases our confidence in the relevance of some interactions which we initially included. We implemented intuitive consistency measures for all possible subnetworks (Figure 1C) and for selected paths (Figure 1D) that include PEBP1 as the target node.

In addition to filtering out irrelevant and insignificant interactions, we used the knockdown dataset to identify regulatory links that were not previously reported in the literature. We inferred potential edges from significant changes in gene expression as a result of the knockdown of a given node (Figure 2). These links would augment the model by increasing the number of edges and mitigating biases arising from tendencies in the literature toward studying certain pathways and specific disease contexts. We suspected that larger effect sizes (perturbation amplitudes) could bias the agreement measures. However, neither in the case of subnetwork concordance nor the path coherence was this bias big enough to influence the results (Figure 5C and Figure 6C).

The value of any predictive model is to make testable hypotheses. Here, we suggested two key pathways regulating PEBP1 (Figure 7). First, the inhibition of the zinc transcriptional repressor (SNAI1) results in PEBP1 suppression [75]. In contrast to PEBP1’s role in metastasis, SNAI1 induced breast cancer EMT and metastasis by directly repressing the epithelial markers E-cadherin (CDH1) [74]. Martin and colleagues found SNAI1 to be overexpressed in invasive metastatic breast cancer compared to normal breast tissue [93]. Moreover, RELA (P65), which is one of five subunits of the Nuclear factor-kappaB family, positively regulated SNAI1 transcription. NF-κB binds directly to the SNAI1 promoter, resulting in increased SNAI1 transcription [94]. Overall, NF-κB, SNAI1, and PEBP1 form a feedback loop that regulates the development of metastasis and resistance to apoptosis [95]. MTA3 is another upstream node of SNAI1 and suppresses its transcription. Similarly, MTA3 represses the transcription of a series of EMT-promoting genes such as ZEB2 and N-cadherin [81]. Our model is consistent with the above-mentioned studies.

Another pathway to regulate PEBP1 in our model is through ESR1 and NME1. NME1 dramatically inhibits the NF-κB signaling pathway and alters the transcription of metastasis-related genes, reducing metastatic pulmonary cells [96]. Additionally, NME1 regulates gene expression in breast cancer cells. Moreover, NME1-dependent genes have a prognostic value as they predicted survival in breast cancer patients [50]. Here, we propose for the first time an alternative link between the ESR1 and PEBP1 induction via the kinase NME1.

We experimentally tested the effect of several drugs on PEBP1/RKIP and its regulators. In addition to verifying the predicted interactions, we wanted to highlight the uncertainty associated with this model. The model had three points of uncertainty. The first is whether a given drug produces the predicted effect on the terminal node of interest. Four out of the six tested drugs conformed to the expectations. The second uncertainty is whether that effect is achieved through both, either or a third unspecified pathway. Sorafenib inhibited PEBP1 by activating the inhibitory pathway of RELA/SNAI1 and inhibiting the other activation pathway of ESR1/NME. By contrast, Cisplatin and Imatinib repressed PEBP1 through only one of the two pathways. The third uncertainty stems from the fact that the farther back in the pathway we travel, the less accurate the predictions are. For example, Epirubicin activated PEBP1 by inducing NME1, but the effect on ESR1 was not as expected.

The validity of this model is limited to how much the included interactions represent the underlying biology. Another limitation is the amount of gene expression signal captured by the knockdown and perturbation data. We carefully curated existing studies to have as many and as accurate as possible links between metastasis suppressors. This model is guaranteed to evolve with the rapidly accumulating biomedical literature. As there is no existing high-throughput method to measure the activity of diverse biological entities in parallel, we relied on a reverse engineering approach to infer the activity from gene expression changes in their downstream nodes. The reverse reasoning approach assumes that the changes in gene expression of a group of nodes reflect the activation or inhibition of their upstream proteins.

The regulatory links in the final model were derived from the literature and prioritized by our analysis. However, further validation of these interactions is necessary first to test their relevance and the reliability of this approach. We supported our observations by devising two notions of consistency, concordance, and coherence. The first of the two shows the general agreement between expectations and observations in the subgraphs and the response to drug treatment. On the other hand, coherence shows consistency in particular pathways and can be thought of as a measure of the relevance of specific links. We elected to use those two measures of agreement for ease of interpretation rather than more sophisticated notions of consistency. We carried out experimental validation and found that our model is most reliable in predicting the drug effect on a target node, but uncertainty increases as to through which pathway and how many upstream nodes are regulated.

## 5. Conclusions

In conclusion, we suggested that the metastasis suppressor PEBP1 is on the receiving end of two key regulatory mechanisms. One inhibitory pathway involves RELA (NFKB p65 subunit) and SNAI1, previously reported to interact with PEBP1. The other pathway involves the estrogen receptor (ESR1), which induces PEBP1 through the kinase NME1. Those two pathways were highly responsive to pharmacological perturbations and would be likely targets for intervention against metastatic breast cancer.

## Figures and Tables

**Figure 1 cancers-13-06098-f001:**
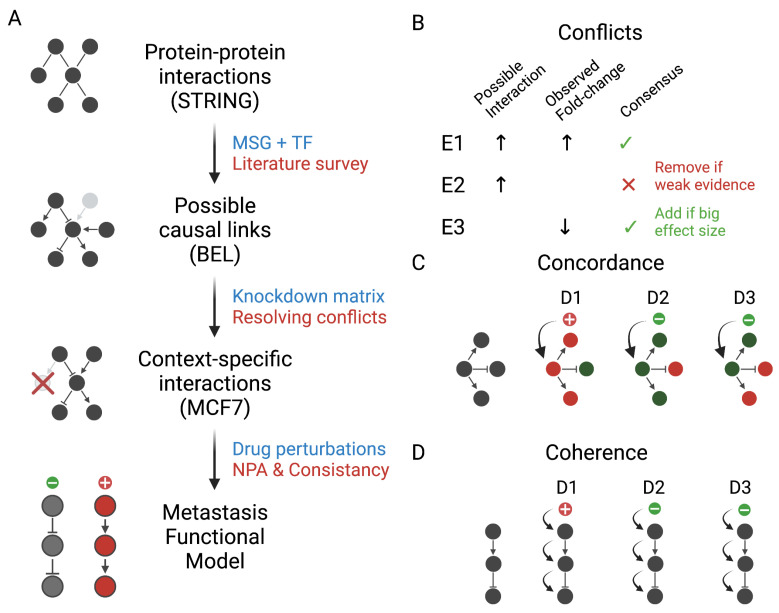
A workflow for building, refining, and evaluating a breast cancer-specific network model of metastasis. (**A**) We queried the String database for protein–protein interactions for metastasis suppressor genes (MSG) and transcription factors (TF). Next, we surveyed the relevant literature to vet and supplement the interactions and express them as possible causal links in the biological expression language (BEL). Finally, we filtered and evaluated the network model using knockdown and drug perturbation datasets. (**B**) Conflicts between the data-driven and curated interactions were resolved by prioritizing the context, the strength of the evidence, and the effect sizes. We scored the metastasis model on drug perturbations using the network perturbation amplitude (NPA). We divided the network into smaller subgraphs and measured the agreement between the expected and the observed direction activity. We considered the consistency in (**C**) subnetworks of nodes connected to an upstream by one edge (concordance) and (**D**) paths connecting a particular node to its upstream by a sequence of edges (coherence).

**Figure 2 cancers-13-06098-f002:**
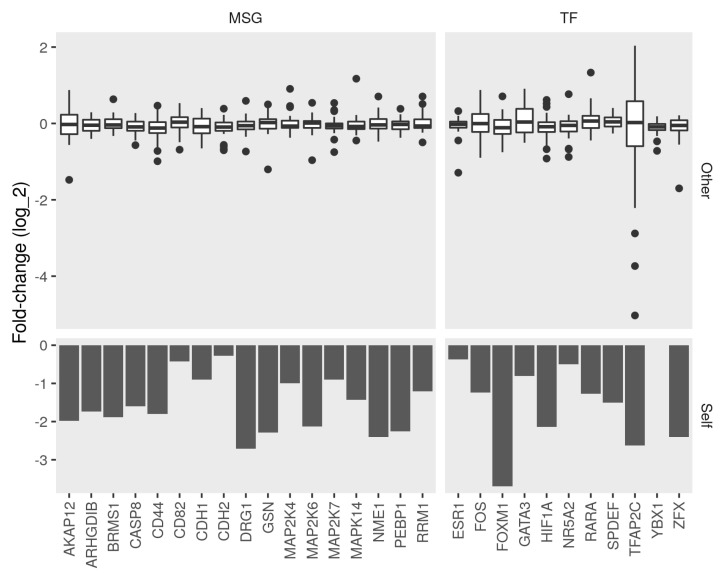
Fold-change between control and knockdown of metastasis suppressors and transcription factors. We obtained expression profiles of seventeen metastasis suppressor genes (MSG) and eleven transcription factor (TF) knockdowns in the MCF7 cell line. We calculated the differential expression between the knockdown and control samples. Bottom, bars represent the fold-change (log${}_{2}$) of the target gene (Self) between the control and knockdown. Top, box plots represent the distribution of the fold-change of (Other) MSGs and TFs in the corresponding knockdown condition.

**Figure 3 cancers-13-06098-f003:**
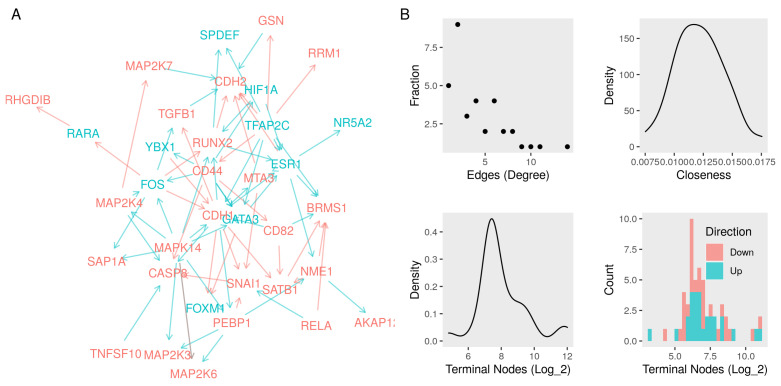
Network of interactions among metastasis suppressors and their regulators. (**A**) A network graph of seventeen metastasis suppressors and eleven transcription factor nodes. We curated the edges (red, repression or blue, activation) from protein–protein interactions, literature survey, and context-specific gene expression data (Functional layer). (**B**) Top left, the fractions of nodes with a given number of edges (degree). Top right, the density function of the sum of the shortest paths between every node and all others (closeness). We built a transcript layer of significant changes in gene expression (absolute log${}_{2}$ fold-change > 0.5 and *p*-value < 0.01) as a result of knocking down every node in the functional layer. Bottom left, the density function of the numbers of nodes in the transcript layer connected to the nodes in the network. Bottom right, a histogram of the numbers up-and down-regulated nodes in the transcript layer.

**Figure 4 cancers-13-06098-f004:**
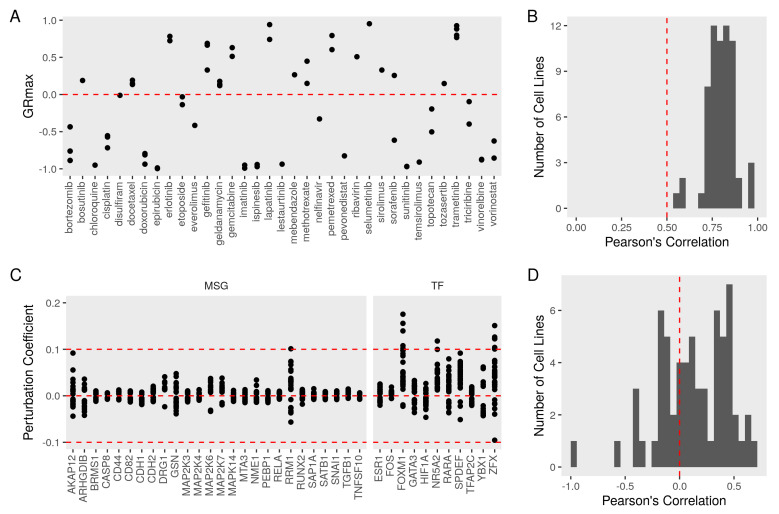
Growth inhibition of MFC7 and metastasis network perturbation under drug treatments. We obtained growth inhibition and gene expression of breast cancer cell lines (*n* = 71) treated with different drugs (*n* = 35). (**A**) Maximum growth rate inhibition (GRmax) of MCF7 from different does and replicates after accounting for the baseline growth rate. Values of 1 indicate no inhibition, while −1 indicates the maximum inhibition of growth. (**B**) A histogram of the Pearson’s correlation coefficients between MCF7 GRmax and other cell lines. (**C**) We computed the network perturbation amplitudes (NPA) for the metastasis network model and every node in the network under drug (*n* = 35) treatment in different cell lines (*n* = 67). Positive values indicate the drug treatment activates the node in MCF7 and negative values indicate repression of the node. (**D**) A histogram of the Pearson’s correlation coefficients between MCF7 NPA and other cell lines.

**Figure 5 cancers-13-06098-f005:**
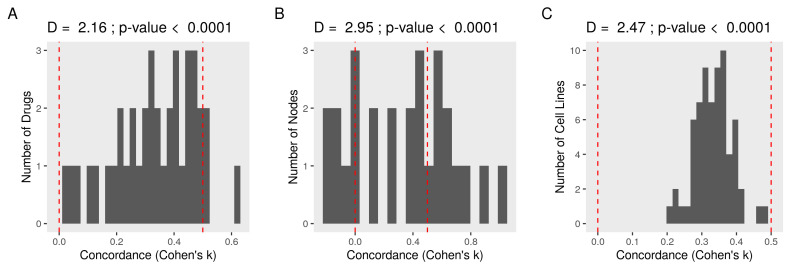
Concordance of expectations and observations in the subnetworks of the metastasis model. We calculated and binarized (1 for activation or −1 for repression) the perturbation coefficients of every node in the network. We then evaluated the agreement between the expected and observed direction of change in the subnetworks of nodes connected to an upstream by one edge (concordance). First, we multiplied the drug’s effect on the upstream node by the sign of the edges to form expectations. Next, we compared the expectations with the actual perturbation coefficients of the corresponding nodes. Negative Cohen’s κ indicates worse and positive better agreement than expected by chance. Finally, we compared the probability distribution of the concordance to randomly generated values using Kolmogorov-Smirnov (KS) test. D${}^{+}$ is the maximum distance between the cumulative distribution functions (ECDF). (**A**) A histogram of the average concordances for every drug. (**B**) A histogram of the average concordances for every node. (**C**) A similar workflow was applied to other cancer cell lines (*n* = 67), and the concordance values averaged by cell lines are shown as a histogram.

**Figure 6 cancers-13-06098-f006:**
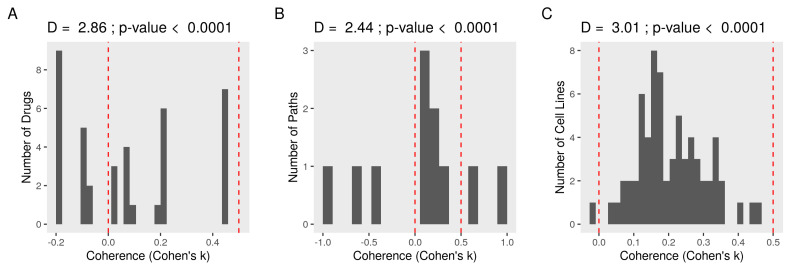
Coherence of expectations and observations in the paths to PEBP1. We calculated and binarized (1 for activation or −1 for repression) the perturbation coefficients of every node in the network. We then evaluated the agreement between the expected and observed direction of change in the paths connecting a PEBP1 to its upstream by a sequence of edges (coherence). First, we multiplied the drug’s effect on the upstream nodes by the sign of the edge connecting it to the next node to form expectations. Next, we compared the expectations with the actual perturbation coefficients of the corresponding nodes. Negative Cohen’s κ indicates worse and positive better agreement than expected by chance. Finally, we compared the probability distribution of the coherence to randomly generated values using Kolmogorov–Smirnov (KS) test. D${}^{+}$ is the maximum distance between the cumulative distribution functions (ECDF). (**A**) A histogram of the average coherence for every drug. (**B**) A histogram of the average coherence for every path. (**C**) A similar workflow was applied to other cancer cell lines (*n* = 67), and the coherence values averaged by cell line are shown as a histogram.

**Figure 7 cancers-13-06098-f007:**
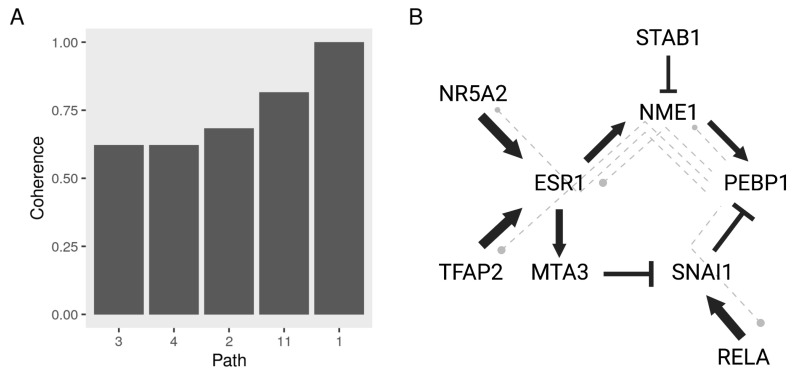
A functional model of PEBP1 interactions with other metastasis suppressors and regulators. We calculated the coherence of the paths leading to PEBP1 in the metastasis networks the expected to the observed perturbations with various drug treatments. (**A**) Top five paths based on the average coherence across drug treatments. (**B**) A network graph of the top coherent PEBP1 paths. Nodes represent biological entities and connect to each other by activation (arrow) or repression (blunt) edges. The thickness of the edges indicates the coherence of the path averaged by drugs.

**Figure 8 cancers-13-06098-f008:**
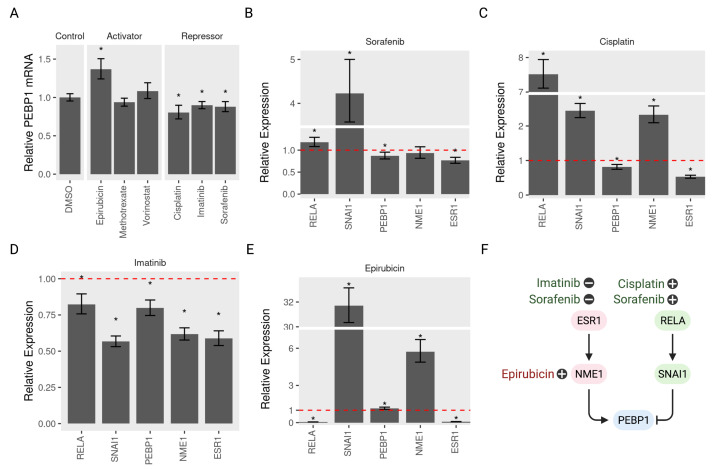
Activation and repression of PEBP1/RKIP and its regulatory pathways. MCF7 cells (*n* > 5) were treated with three activators (Epirubicin, Methotrexate, or Vorinostat), three repressors (Cisplatin, Imatinib, or Sorafenib), or the control DMSO for 24 h at 1.5 μM dose. Total RNA was extracted and amplified using PCR. The amount of five mRNA (RELA, SNAI1, PEBP1, NME1, and ESR1) was quantified relative to the control GAPDH using RT-qPCR. (**A**) The relative RKIP mRNA with all six drug treatments compared to the control. The relative amount of five mRNA in three repressors treatments (**B**) Sorafenib, (**C**) Cisplatin or (**D**) Imatinib, and one activator (**E**) Epirubicin. (**F**) A model of the pathways regulating RKIP and the potential activators and repressors. Drugs are positioned near the node they regulate with plus or minus signs to indicate activation or repression. * indicates *p* < 0.05.

**Table 1 cancers-13-06098-t001:** Metastasis suppressor genes.

Category	Genes
Cell-cell adhesion	Epican (CD44), Tetraspanin 27 (CD82), Cadherin 11 (CDH11), Cadherin 2 (CDH2), Cadherin 1 (CDH1) and Gelsolin (GSN)
Scaffolding	Gravin/a-kinase anchor protein 12 (AKAP12)
MAPK	Dual specificity mitogen-activated protein kinase kinase 6 (MAP2K6), 4 (MAP2K4), 7 (MAP2K7) and Mitogen-activated protein kinase 14 (MAPK14)
Transcription	NME/NM23 Nucleoside Diphosphate Kinase 1 (NME1) and breast cancer metastasis-suppressor (BRMS1)
GTP-binding	Rho GDP Dissociation Inhibitor Beta (ARGHDIB) and Developmentally-regulated GTP-biding protein 1 (DRG1)
Other	Ribonucleotide Reductase Catalytic Subunit M1 (RRM1) and Phosphatidylethanolamine-binding protein 1 (PEBP1)

**Table 2 cancers-13-06098-t002:** Transcription factors targeting metastasis suppressor genes in MCF7.

TF	Name	Dataset ID	Ref.
ESR1	Estrogen receptor 1	GSE10061	[17]
FOS	Fos Proto-Oncogene AP-1 Transcription Factor Subunit	GSE36586	[18]
FOXM1	Forkhead Box M1	GSE55204	[19]
GATA3	GATA Binding Protein 3	GSE39623	[20]
HIF1A	Hypoxia Inducible Factor 1 Subunit Alpha	GSE3188	[21]
NR5A2	Nuclear Receptor Subfamily 5 Group A Member 2	GSE47803	[22]
RARA	Retinoic Acid Receptor Alpha	GSE26298	[23]
SPDEF	SAM Pointed Domain Containing ETS Transcription Factor	GSE40985	[24]
TFAP2C	Transcription Factor AP-2 Gamma	GSE26740	[25]
YBX1	Y-Box Binding Protein 1	GSE28433	[26]
ZFX	Zinc Finger Protein X-Linked	ENCSR005AHI	[27]

**Table 4 cancers-13-06098-t004:** RT-qPCR primers.

Gene	Forward Primer	Reverse Primer
ESR1	5′-TGGAGTCTGGTCCTGTGAGG-3′	5′-GGTCTTTTCGTATCCCACCTTTC-3′
SNAI1	5′-CCAGTGCCTCGACCACTATG-3′	5′-CTGCTGGAAGGTAAACTCTGG-3′
RELA	5′-CCTATAGAAGAGCAGCGTGGG-3′	5′-AGATCTTGAGCTCGGCAGTG-3′
NME1	5′-ACTAAGTCAGCCTGGTGTGC-3′	5′-CGCCTTGAAAGACGATCCCT-3′
PEBP1	5′-GTCACACTTTAGCGGCCTGT-3′	5′-CTCTCCGATTATGTGGGCTC-3′
GAPDH	5′-TGCACCACCAACTGCTTAGC-3′	5′-GGCATGGACTGTGGTCATGAG-3′

## Data Availability

Data are contained within the article.

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
