# Peer review of "A Functional Network Model of the Metastasis Suppressor PEBP1/RKIP and Its Regulators in Breast Cancer Cells"

_cancers, 2021, doi:10.3390/cancers13236098_

Round 1
Reviewer 1 Report
The authors describe an approach of characterising the pathways involved in the regulation of the metastasis suppressor PEBP1. Using text mining, a gene knockdown dataset in the breast cancer cell line MCF7 and drug inhibition data, two regulatory mechanisms were identified involving RELA/SNAI1 and ESR1/NME1.
The paper is well written with a few minor grammatical issues
Line 168: the authors state that additional interactions were added to their model that were not originally identified in the protein-protein interaction model from the STRING database. Why were these added? What was the justification for these additions?
Figure 2: There are 17 MSGs in the figure but throughout the manuscript it states 18.
Figure 2: It is not very clear to the reader what the top and bottom sections of this figure represent. It is not clear if the box plot and the bar chart are for the same genes indicated or are different. The needs to be clearer. Perhaps separate with labels A and B
Figure 8F: I don’t think this pathway represents a clear summary of the regulators from the mRNA. Specifically, why is sorafenib shown twice on the pathway while the other 3 drugs also decrease expression of other regulators in the pathway? I would suggest that the authors expand on this a bit further.
Discussion: Could the model be expanded to include post-translational modifications that would affect the pathways for protein activation? How was this improve your predictions for targeted therapy?
Does the use of MCF7 cell line take into account any mutations in the cell line that may impact the expression of PEBP1?
Author Response
The authors describe an approach of characterising the pathways involved in the regulation of the metastasis suppressor PEBP1. Using text mining, a gene knockdown dataset in the breast cancer cell line MCF7 and drug inhibition data, two regulatory mechanisms were identified involving RELA/SNAI1 and ESR1/NME1.
The paper is well written with a few minor grammatical issues
- Line 168: the authors state that additional interactions were added to their model that were not originally identified in the protein-protein interaction model from the STRING database. Why were these added? What was the justification for these additions?
We used a simple procedure to resolve conflicts between the literature search and the observed changes in response to the nodes' knockdown (Figure 1B). In some cases, including the one referred to in the question, the knockdown of one node resulted in a large change in the expression of another. In cases like that, we added an edge between the two nodes to represent a potential interaction. Adding additional interaction can help improve network accuracy. the precise of the network.
- Figure 2: There are 17 MSGs in the figure but throughout the manuscript it states 18.
We corrected the text to state that 17 MSG were included in the study.
- Figure 2: It is not very clear to the reader what the top and bottom sections of this figure represent. It is not clear if the box plot and the bar chart are for the same genes indicated or are different. The needs to be clearer. Perhaps separate with labels A and B
We modified Figure 2 legend to state that the bottom section of the figure shows the effect of the knockdown on the target gene, and the top section showed the effect of the knockdown of that gene on other MSGs or TFs.
- Figure 8F: I don’t think this pathway represents a clear summary of the regulators from the mRNA. Specifically, why is sorafenib shown twice on the pathway while the other 3 drugs also decrease expression of other regulators in the pathway? I would suggest that the authors expand on this a bit further.
We modified Figure 8 legend to state that the F panel shows the predicted pathways as well as the verified drug activators and repressors. The position of each drug indicates the node it affects. The plus and minus circles indicate the direction of the effect. Sorafenib seems to regulate PEBP1 through either pathway, hence indicated twice. It could inhibit the ESR1/NME1 or induce RELA/SNAI1.
- Discussion: Could the model be expanded to include post-translational modifications that would affect the pathways for protein activation? How was this improve your predictions for targeted therapy?
The network perturbation amplitude (NPA) method uses gene expression data to predict the function of the nodes in the causal network. Some of the causal links in the network may include the modification of the downstream genes post-transcriptionally. However, in the case study, upstream nodes are likely to induce or repress the transcription of PEBP1. It would be interesting to supplement this methodology with post-transcriptional modification in future research. One possibility would be to explain the effect sizes of the drug treatments another would be to explain some of the output of verification that did not agree with the predicitons.
- Does the use of MCF7 cell line take into account any mutations in the cell line that may impact the expression of PEBP1?
Our model is based on gene expression data only, therefore it doesn't take mutations in the cell line into account.

Reviewer 2 Report
The authors described a reverse reasoning approach methodology for predicting PEBP1 regulation pathways in breast cancer model, which could be useful to continue studies on the choice of therapeutic targets. However, the manuscript should be improved to be consider for publication, since some lacks of information are detected. Please note that the following contributions or corrections are solely for the purpose of improving the quality of the work presented:
In the main text, the authors say in several times they used the eighteen genes included in Table 1. And in the Table 1 are only 17 genes. Please, check if these a lack of a gene in the table or correct the number in the main text in the corresponding sites (line 58, 65, 105, 119, legend figure 3...)
In mat&met section, in the point 2.3 the authors specified they used 35 drugs in 71 breast cancer cell lines. It seems that they used all of them and not a dataset. Then in the results they specified but it is confused, so, please, include in the point 2.3 an explanation such as in the line 231.
The legend of the Figure 1 contains duplicated information from the main text from results’ point 3.1. Remove it and rewrite the legend appropriately. Review all the figures legends and remove the methodology information. Consider to include it in the material and methods section.
The results’ point 3.2 is duplicated from 3.1. Unify them.
Line 262: duplicated sentence 'was very weak'. Delete one.
Line 275: The phrase is not understood, maybe the word 'however' should be removed.
Regarding the validation results:
- Figure 8A shows the results of the expression of PEBP1 after having subjected the MCF7 cells to different treatments. In the main text, the authors mention that 4 of the 6 drugs tested gave rise to the expected changes. However, the graph shows that only three drugs produced significant changes in PEBP1 levels: epirubicin, imatinib and sorafenib. Why do the authors use cisplatin instead of imatinib for validation?
- Figure 8B: Sorafenib treatment induces significant changes in the expression of RELA, SNAI and ESR1, but not in NME1. Specify in the main text
- Figure 8C: The data on the expression of PEBP1 contradict what appears in figure A.
- Figure 8D: Why do the authors analyze the expression of the genes of interest if the expression of PEBP1 does not change after treatment with vorinostat? The result is repeated in both Figures A and D, not stastitical differences are observed.
- Figure 8E: The expression of PEBP1 does not appear to have a significant change despite the asterisk present in the graph. Furthermore, the expression of RELA decreases while that of SNAIL increases. In the same way, the expression of NME1 increases while that of ESR1 decreases. The authors mention: the expression of PEBP1 is induced by the expression of NME1 or ESR1. A high expression of SNAIL1 should repress the expression of PEBP1 and a high expression of NME1 an induction of the expression of PEBP1. However, in the absence of ESR1, this protein cannot induce the expression of PEBP1. The results shown in figure E do not correspond to the text description or to the proposed action scheme.
- Figure 8F: The activator and repressor scheme generates confusión.
Discussión: Paragraph 3 is duplicated from the results, abbreviate it. Paragraph 4 described a regulation loop previously described in the literature (ref 74, 89…). In general, the discussion is poorly argued, the results are repeated and little reflection is provided based on the existing literature. In addition, as a conclusion of the work, the authors talk about a regulatory path for PEBP1 that was already described, which is not new.
Author Response
Reviewer #2
The authors described a reverse reasoning approach methodology for predicting PEBP1 regulation pathways in breast cancer model, which could be useful to continue studies on the choice of therapeutic targets. However, the manuscript should be improved to be consider for publication, since some lacks of information are detected. Please note that the following contributions or corrections are solely for the purpose of improving the quality of the work presented:
- In the main text, the authors say in several times they used the eighteen genes included in Table 1. And in the Table 1 are only 17 genes. Please, check if these a lack of a gene in the table or correct the number in the main text in the corresponding sites (line 58, 65, 105, 119, legend figure 3...)
We corrected the text to state that 17 MSG were included in the study.
- In mat&met section, in the point 2.3 the authors specified they used 35 drugs in 71 breast cancer cell lines. It seems that they used all of them and not a dataset. Then in the results they specified but it is confused, so, please, include in the point 2.3 an explanation such as in the line 231.
We revised the section to explain the sources of perturbation data and how they were used in the analysis.
- The legend of the Figure 1 contains duplicated information from the main text from results’ point 3.1. Remove it and rewrite the legend appropriately. Review all the figures legends and remove the methodology information. Consider to include it in the material and methods section.
We intentionally wrote the figure legends to be self-contained and provided some methodological details to make it easier to read each figure on its own. Some overlap between the main text and the figure legend are unavoidable
- The results’ point 3.2 is duplicated from 3.1. Unify them.
We revised the text in 3.2 to avoid repetition. The first subsection of the Results describes the full study workflow and each subsection after that deals with one of the workflow steps so there might be some expected overlap in the text. We renamed the first subsection to indicate that it is intended as a description of the the workflow for generating the network which we consider as an essential part of the results.
- Line 262: duplicated sentence 'was very weak'. Delete one.
We deleted the repeated phrase.
- Line 275: The phrase is not understood, maybe the word 'however' should be removed.
We revised the sentence to clarify the meaning.
- Regarding the validation results:
- Figure 8A shows the results of the expression of PEBP1 after having subjected the MCF7 cells to different treatments. In the main text, the authors mention that 4 of the 6 drugs tested gave rise to the expected changes. However, the graph shows that only three drugs produced significant changes in PEBP1 levels: epirubicin, imatinib and sorafenib. Why do the authors use cisplatin instead of imatinib for validation?
After repeating the experiment shown in Figure 8A, we found three consistent repressors (Sorafenib, Cisplatin and Imatinib) and one activator (Epirubicin). The updated figure excluded the Vorinostat data and included the data for Imatinib.
- Figure 8B: Sorafenib treatment induces significant changes in the expression of RELA, SNAI and ESR1, but not in NME1. Specify in the main text
We revised the text to state that the chances in NME1 were not significant at the specified cutoff.
- Figure 8C: The data on the expression of PEBP1 contradict what appears in figure A.
After repeating the experiment shown in Figure 8A, the data is consistent with panel 8C. We believe the reviewer is referring to the inconsistency in the significance of the Cisplatin effect on PEBP1. In the revised figure, Cisplatin repressed PEBP1 consistently (P < 0.05).
- Figure 8D: Why do the authors analyze the expression of the genes of interest if the expression of PEBP1 does not change after treatment with vorinostat? The result is repeated in both Figures A and D, not stastitical differences are observed.
We removed the data of Vorinostat and added new data for Imatinib which was significant and consistent with our expectations.
- Figure 8E: The expression of PEBP1 does not appear to have a significant change despite the asterisk present in the graph. Furthermore, the expression of RELA decreases while that of SNAIL increases. In the same way, the expression of NME1 increases while that of ESR1 decreases. The authors mention: the expression of PEBP1 is induced by the expression of NME1 or ESR1. A high expression of SNAIL1 should repress the expression of PEBP1 and a high expression of NME1 an induction of the expression of PEBP1. However, in the absence of ESR1, this protein cannot induce the expression of PEBP1. The results shown in figure E do not correspond to the text description or to the proposed action scheme.
As with any predictive model, there is uncertainty associated with it. We explain in the Discussion that the data presented in Figure 8 is mainly to highlight these uncertainties and how to use these kinds of predictions. In the case of PEPB1 regulators, there were three different points we needed to worry about. First, do the drugs produce the predicted effect? Second, which of the two pathways, if any, relays the effect on PEBP1? Third, how far back in the pathway does each drug go? Our results indicated that several of the tested drugs conformed to the expected effect on the target node, PEBP1. Drugs regulated PEBP1 through the two pathways (e.g. Sorafenib) or only one pathway (e.g. Cisplatin or Imatinib). Finally, Drugs could regulate two of the upstream nodes of PEBP1 (e.g. Sorafenib) or only one (e.g. Epirubicin). We concluded that our model is most reliable in predicting the drug effect on a target node, but uncertainty increases as to through which pathway and how many upstream nodes are regulated.
- Figure 8F: The activator and repressor scheme generates confusión.
We modified Figure 8 legend to state that the F panel shows the predicted pathways as well as the verified drug activators and repressors. The position of each drug indicates the node it affects. The plus and minus circles indicate the direction of the effect.
- Discussión: Paragraph 3 is duplicated from the results, abbreviate it. Paragraph 4 described a regulation loop previously described in the literature (ref 74, 89…). In general, the discussion is poorly argued, the results are repeated and little reflection is provided based on the existing literature. In addition, as a conclusion of the work, the authors talk about a regulatory path for PEBP1 that was already described, which is not new.
We revised paragraph 3 in the Discussion to avoid repetition. The model we present in this study consist of two regulatory pathways of PEPB1 and the drugs that manipulate them. One of the two pathways (RELA/SNAI1) was previously reported in the literature, the other was not (ESR1/NME1). In addition, we suggest some of the drugs that target PEBP1 through either. We revised the discussion section to reflect on these findings and highlight the advantages and shortcomings of this model.

Reviewer 3 Report
The present manuscript describes the use of the reverse causal reasoning approach, to build a functional model of metastasisn in breast cancer. Furthermore, authors aimed at evaluating for a possible cell growth inhibition drug treatment. They investigated the metastasis suppressor PEBP1/RKIP associated with two significant regulatory mechanisms (one involving RELA and SNAI1, and the other involving ESR1), together with the validation of some of the regulatory links in MCF7 cell line. In my opinion the study looks interesting, brings novelty in the potential use of big datahigh-throughput technologies and does provides some interesting data. There are however, few clarifications and modification required before further consideration.
Questions and concerns to be addressed properly:
1. Introduction:
- The term "Reverse causal reasoning" needs to be described more extensively. It is not explained clearly enough in two sentences.
- There is no reference when talking about metastasis suppressor genes.
2. Methodology:
- Maybe I am wrong, but I am not able to count more than 17 genes in table 1, and the authors mention 18 genes in section 2.1 and 2.2 related to the table.
- In reference 23, from Marino et al (2014), they study 19 MSGs, why do the authors only refer to 17?
- The fact that the tables are not cited in the text following the correct order is strange. Table 4 is cited in section 2.1 and table 3 is not cited until section 2.8. In my opinion the order of tables needs to be changed.
- Which are the FDR and pvalue levels applicated when using bioinformatic data?
3. Results:
- The first section of the results is very similar to methodology section and it makes that section very repetitive.
- Maybe I am wrong, but I am not able to count more than 17 genes in figure 2, and the authors mention 18 genes in section 3.2 related to the figure.
- In section 3.2 the table 4 needs to be cited, as the authors refer to those results " the interactions where coded in the biological expression language and were used as the functional layer for the network perturbation amplitude analysis".
- In section 3.6, authors comment that "they selected several drugs that activate or repress PEBP1", how many? Why some of them work and others did not work? It could be discussed in discussion section.
4. Discussion:
- In the most part, discussion reads more like an extension of Results. This section should be enriched by comparing the obtained results with similar works and not just a repetition of the obtained results.
- About the "Reverse causal reasoning" the authors add only one reference to this methodology. Is it robust enough to believe in the obtained results? I would appreciate more discussion about the use of this methodology in science.
- Why is the reason for selecting PEBP1/RKIP for the final part of the experimentation? There is nothing explained about that neither in introduction when first emerged, nor in results or discussion, when the obtained results are described. Similar results are expected with other individual MSGs?
5. General:
- The authors need to explain the terms or achronims only when first emerge, and not in all sections. For example metastasis suppresor genes (MSG) emerges first in line 38, but it is explain again in line 164 and in some figure leyends. The same happends with Network perturbation amplitude (NPA), it emerges first in line 90, but it is explain again in line 91, 183, 238, and in some figure leyends.
Author Response
Reviewer #3
The present manuscript describes the use of the reverse causal reasoning approach, to build a functional model of metastasisn in breast cancer. Furthermore, authors aimed at evaluating for a possible cell growth inhibition drug treatment. They investigated the metastasis suppressor PEBP1/RKIP associated with two significant regulatory mechanisms (one involving RELA and SNAI1, and the other involving ESR1), together with the validation of some of the regulatory links in MCF7 cell line. In my opinion the study looks interesting, brings novelty in the potential use of big datahigh-throughput technologies and does provides some interesting data. There are however, few clarifications and modification required before further consideration.
Questions and concerns to be addressed properly:
- Introduction:
- The term "Reverse causal reasoning" needs to be described more extensively. It is not explained clearly enough in two sentences.
We revised the introduction section to describe the reverse causal reasoning approach.
- There is no reference when talking about metastasis suppressor genes.
We add a reference to the sentence talking about metastasis suppressor genes in the introduction.
- Methodology:
- Maybe I am wrong, but I am not able to count more than 17 genes in table 1, and the authors mention 18 genes in section 2.1 and 2.2 related to the table.
The correct number of genes is 17. We corrected the text to reflect this fact.
- In reference 23, from Marino et al (2014), they study 19 MSGs, why do the authors only refer to 17?
We excluded from this study two genes. The knockdown of the two genes was apparently not successful. We couldn't confirm the change in the corresponding genes (targets) as a result of the knockdown and so excluded them.
- The fact that the tables are not cited in the text following the correct order is strange. Table 4 is cited in section 2.1 and table 3 is not cited until section 2.8. In my opinion the order of tables needs to be changed.
We modified the position of the tables to reflect the order in which they were referenced in the text.
- Which are the FDR and pvalue levels applicated when using bioinformatic data?
We used the FDR values when calculating the fold-change and supplied them as input to the NPA method. NPA doesn't require a specific cutoff but takes the FDR to indicate the confidence level for each gene.
- Results:
- The first section of the results is very similar to methodology section and it makes that section very repetitive.
The first subsection of the Results describes the full study workflow and each subsection after that deals with one of the workflow steps so there might be some expected overlap in the text.
- Maybe I am wrong, but I am not able to count more than 17 genes in figure 2, and the authors mention 18 genes in section 3.2 related to the figure.
The correct number of genes is 17. We corrected the text to reflect this fact.
- In section 3.2 the table 4 needs to be cited, as the authors refer to those results " the interactions where coded in the biological expression language and were used as the functional layer for the network perturbation amplitude analysis".
We cited table 4 in section 3.2.
- In section 3.6, authors comment that "they selected several drugs that activate or repress PEBP1", how many? Why some of them work and others did not work? It could be discussed in discussion section.
The output of our approach includes a perturbation value and statistics for every drug. Most drugs produce a negligible effect on the target of interest. We chose to test 6 drugs that were effective in perturbing PEBP1. We modified the text of the discussion to refer to this point. Paragraphs 2-4 in the discussion, after modification, address the issue. We discuss reasons why some of the predictions are expected to be less accurate.
- Discussion:
- In the most part, discussion reads more like an extension of Results. This section should be enriched by comparing the obtained results with similar works and not just a repetition of the obtained results.
We modified the text of the discussion section to address this issue.
- About the "Reverse causal reasoning" the authors add only one reference to this methodology. Is it robust enough to believe in the obtained results? I would appreciate more discussion about the use of this methodology in science.
We added a further discussion to the second paragraph of the discussion to show the previous use of the approach in the literature.
- Why is the reason for selecting PEBP1/RKIP for the final part of the experimentation? There is nothing explained about that neither in introduction when first emerged, nor in results or discussion, when the obtained results are described. Similar results are expected with other individual MSGs?
We expect our approach to work as well for any node in the network. We presented the regulation of PEBP1 as a case study to highlight the advantages and disadvantages of the approach. PEBP1 happened to be a metastasis suppressor that our lab is interested in.
- General:
- The authors need to explain the terms or achronims only when first emerge, and not in all sections. For example metastasis suppresor genes (MSG) emerges first in line 38, but it is explain again in line 164 and in some figure leyends. The same happends with Network perturbation amplitude (NPA), it emerges first in line 90, but it is explain again in line 91, 183, 238, and in some figure leyends.
We modified the text to address this issue. Figure legends were written to be self-contained so we elected to define the key acronyms in full when possible.

Round 2
Reviewer 1 Report
I am satisfied that the authors have addressed all of my concerns.
Reviewer 2 Report
Thanks to the authors for taking the suggestions into account. This new version is much better understood and I consider it suitable for publication.
Reviewer 3 Report
In my opinion authors have responded adequately to all my comments and the manuscript too has been revised reasonably well. I do not have any further concerns.